# Current Evidence on Vasa Previa without Velamentous Cord Insertion or Placental Morphological Anomalies (Type III Vasa Previa): Systematic Review and Meta-Analysis

**DOI:** 10.3390/biomedicines11010152

**Published:** 2023-01-07

**Authors:** Yuki Takemoto, Shinya Matsuzaki, Satoko Matsuzaki, Mamoru Kakuda, Misooja Lee, Harue Hayashida, Michihide Maeda, Shoji Kamiura

**Affiliations:** 1Department of Gynecology, Osaka International Cancer Institute, Osaka 541-8567, Japan; 2Department of Obstetrics and Gynecology, Osaka University Graduate School of Medicine, Osaka 565-0871, Japan; 3Department of Obstetrics and Gynecology, Osaka General Medical Center, Osaka 558-8558, Japan; 4Department of Forensic Medicine, School of Medicine, Kindai University, Osaka 589-8511, Japan

**Keywords:** vasa previa, type III vasa previa, assisted reproductive technique, bilobed placenta, succenturiate lobes, accessory lobes, velamentous cord insertion

## Abstract

Vasa previa carries a high risk of severe fetal morbidity and mortality due to fetal hemorrhage caused by damage to unprotected fetal cord vessels upon membrane rupture. Vasa previa is generally classified into types I and II. However, some cases are difficult to classify, and some studies have proposed a type III classification. This study aimed to review the current evidence on type III vasa previa. A systematic literature search was conducted, and 11 articles (2011–2022) were included. A systematic review showed that type III vasa previa accounts for 5.7% of vasa previa cases. Thirteen women with type III vasa previa were examined at a patient-level analysis. The median age was 35 (interquartile range [IQR] 31.5–38) years, and approximately 45% were assisted reproductive technology (ART) pregnancies. The median gestational week of delivery was 36 (IQR 34–37) weeks; the antenatal detection rate was 84.6%, and no cases reported neonatal death. The characteristics and obstetric outcomes (rate of ART, antenatal diagnosis, emergent cesarean delivery, gestational age at delivery, and neonatal mortality) were compared between types I and III vasa previa, and all outcomes of interest were similar. The current evidence on type III vasa previa is scanty, and further studies are warranted.

## 1. Introduction

Vasa previa (VP) is a high-risk obstetric disease characterized by unprotected fetal vessels running near or across the internal cervical os [1,2,3,4,5,6]. The distance between unprotected fetal vessels and internal cervical os for adverse fetal outcomes is debatable [7], and several authors have recognized that fetal blood vessels running near (within 2 cm) or across the internal cervical os are VP [8,9,10,11,12,13,14]. The estimated prevalence of VP is approximately 0.05%, and the risk factors of VP include velamentous cord insertion (VCI), low-lying placenta, accessory lobe of the placenta, assisted reproductive technology (ART), and twin pregnancy [2,15,16,17]. Since ART correlates with a higher rate of VCI and low-lying placenta, ART pregnancy is associated with a higher incidence of VCI than non-ART pregnancy [2].

A significant clinical problem of VP is an extremely high fetal mortality rate, which is as high as 44% in women with undiagnosed VP. In contrast, the fetal mortality rate was 3% in women with antenatally diagnosed VP [18,19]. In addition, there is a need to identify women with VP often presenting complications with abnormal placentation (low-lying placenta and placenta previa), the risk of postpartum hemorrhage, blood transfusion, and postoperative infection after cesarean delivery, especially in low to middle-income countries (LMIC) [20,21,22,23,24,25,26]. Therefore, antenatal diagnosis of VP is essential, and knowledge of the risk factors and an overview of this rare disease may contribute to improved fetal outcomes. The classification into types I and II VP according to the etiology of VP was proposed by Catanzarite V et al. in 2001, and various studies have used this classification [27,28,29,30,31,32,33,34].

The combination of a low-lying placenta and VCI has been considered a high-risk obstetric condition of type I VP. Moreover, if the velamentous cord vessels run near or across the internal cervical os, these conditions are considered as type I VP [16,35]. In the case of a placenta with a succenturiate lobe or bilobed placenta, unprotected vessels connecting the lobes run within 2 cm of the internal cervical os, which is classified as type II VP [3,27,36,37]. Most VP cases can be classified as type I or type II VP; however, some cases are difficult to classify as either type I or type II. For instance, as shown in Figure 1, the cases of women with unprotected fetal vessels overlying the cervix without either VCI or succenturiate lobes are difficult to classify as either type I or type II. In 2016, Catanzarite V et al. proposed the classification of type III VP [38], and several cases have been reported [5,39,40,41,42,43].

While VP is a well-known fetal life-threatening disease, the risk factors, characteristics, prevalence, obstetric outcomes, and fetal outcomes according to the type of VP have been understudied. A recent our systematic review focused on the differences in obstetric and fetal outcomes between types I and II VP, which comprise the majority of cases. A systematic review found that type II VP comprised approximately 20% of VP, and the obstetric and fetal outcomes were similar between the groups [44]. There has not been a systematic review of the outcomes of type III VP, unlike type I and II VP. Therefore, this study aims to review type III VP to improve the outcomes of this high-risk condition and to identify potential risk factors.

## 2. Materials and Methods

### 2.1. Approach for Systematic Review

A systematic review (PROSPERO registration ID: CRD42022381135) was conducted to determine the outcomes of interest in patients with type III VP. This study is an auxiliary project to our recent systematic review of type II VP [44]. The present study aimed to determine these outcomes of primary interest: (i) obstetric and fetal outcomes of type III VP and (ii) the differences in characteristics and obstetric outcomes between type III VP and other types of VP. The outcome of secondary interest was the estimated prevalence of type III VP.

The present study was conducted according to the guidelines of the systematic review published in 2020 (Preferred Reporting Items for Systematic Reviews and Meta-Analyses statement) [45]. Using three electronic search engines (PubMed [https://pubmed.ncbi.nlm.nih.gov/ (accessed on 5 December 2022)], the Cochrane Central Register of Controlled Trials [https://www.cochranelibrary.com/central/about-central (accessed on 5 December 2022)], and Scopus [https://www.elsevier.com/ja-jp/solutions/scopus (accessed on 5 December 2022)]), a systematic literature search was performed, as reported previously [46,47], with some modifications. Previous studies published before 31 October 2022, were searched and screened using words related to VP. Medical Subject Headings (vasa previa [MeSH]) were also used in PubMed and Cochrane searches.

### 2.2. The Definition of Type I, II, and III Vasa Previa

In this study, the definition of the type of VP was based on previous studies with some modifications [38,48,49]. VP is defined as unprotected fetal vessels running across or near (within 2 cm of internal cervical os) the internal cervical os (Figure 2): (i) Type I: in the case of VCI, the velamentous cord vessels were diagnosed as VP. (ii) Type II: multilobed placenta or placenta with a succenturiate lobe and connecting vessels between the two lobes was diagnosed as VP. (iii) Type III: vessels near or over the internal os with no VCI, multilobed placenta, or placenta with a succenturiate lobe. (iv) If a woman had multiple types of unprotected cord vessels (e.g., VCI and unprotected fetal vessels outside the placenta), the type was classified according to which vessels were diagnosed as VP.

### 2.3. Search Strategy

Using the method described in our previous studies [46,50], articles were screened by reviewing the titles and abstracts of the eligible studies. Y.T., Sa.M., and Sh.M. screened the titles and abstracts to find studies examining the present study’s outcomes of interest.

### 2.4. Analysis of Outcome Measures

The primary aim of this study was to examine the obstetric and fetal outcomes of women with type III VP. Since understanding the difference between type III and other types of VP may be helpful, the characteristics and obstetric outcomes of type III VP patients and other types of VP patients were examined as co-primary outcomes. The secondary outcome was to estimate the prevalence of type III VP.

### 2.5. Selection of Previous Studies

The following inclusion criteria were defined to examine the primary and secondary aims of this study: (1) Comparative studies (type III VP versus type I VP, or type II VP, or women without VP), (2) the prevalence of type III VP was clarified, and (3) a case report regarding type III VP. Among the eligible studies, comparative studies examining the primary and secondary outcomes of present study were further determined.

Previous studies with following criteria were excluded: (1) unknown number of women with type III VP; (2) abstracts not listed in the search engines; (3) not written in English; and (4) narrative reviews, systematic reviews, meta-analyses, conference abstracts, and editorials.

### 2.6. Data Extraction

Sh.M. extracted information for the examination of the outcome of interest. The leading author’s name, year of study, study location, number of control (type I VP, or type II VP, or women without VP) and experimental groups (women with type III VP), and outcomes of interest were documented. M.L. reviewed and verified the data included in the analysis.

### 2.7. Assessment of Bias Risk

The risk of bias in the eligible comparator studies was determined using the Risk of Bias in Non-randomized Studies-of Interventions tool (ROBINS-I), as reported previously [50,51,52,53], with some modifications.

### 2.8. Meta-Analysis Plan

A statistical meta-analysis was conducted for continuous values and dichotomous data, and images and figures were created using the RevMan ver. 5.4.1 software (Cochrane Collaboration, Copenhagen, Denmark). Heterogeneity among studies assessing the outcomes of interest was examined using the *I*^2^ statistic, which weighs the proportion of overall variation. As shown in Appendix A [15], fixed- or random-effects analyses were performed depending on the heterogeneity among the studies. RevMan ver. 5.4.1 software was used to calculate the odds ratio (OR) as an effective measure with a 95% confidence interval (CI).

### 2.9. Statistical Analysis

Fisher’s exact test or the chi-squared test were used to examine the differences in patient background between the groups (experimental and control), as appropriate. All statistical analyses were based on two-sided hypotheses, and a *p*-value of <0.05 was considered statistically significant.

## 3. Results

### 3.1. Study Selection

Figure 3 shows the study selection scheme for the systematic literature search. First, 838 studies were included in the screening, and 11 were identified for descriptive analysis [5,38,39,40,41,42,43,48,49,54,55].

### 3.2. Study Characteristics

A summary of the 11 identified studies is presented in Table 1. Of these studies (*n =* 11), all were retrospective, five were original articles, and six were case reports. Among the eligible studies, the year of publication was between 2011 and 2022 [5,38,39,40,41,42,43,48,49,54,55]. No prospective studies or randomized controlled trials were identified. The majority of studies have been reported in Japan (*n* = 8, 72.7%) [5,39,40,42,43,48,49,55], followed by the USA (*n* = 2, 18.2%) [38,41], and China (*n* = 1, 9.1%) [54].

#### Risk of Bias of included Studies

The risk of bias among the comparator studies was performed as shown in Appendix A. Of those (*n* = 2), a potential severe publication bias was observed [48,55].

### 3.3. Primary Outcome: Obstetric and Neonatal Outcomes of Women with Type III Vasa Previa

Eight studies met these criteria, and 13 women with type III VP were included (Table 2) [5,38,39,40,41,42,43,48,55]. Of these cases (*n* = 13), seven women with type III VP were from two original articles, and six cases were from six case reports.

Thirteen women with type III VP were examined in the patient-level analysis (Table 3). The median age was 35 (interquartile range [IQR] 31.5–38) years, and nearly half (53.8%) were aged 35 years or older. Approximately 45% were ART pregnancies; detailed information regarding the type of ART (e.g., frozen embryo transfer) was unavailable. Preterm delivery was reported in 8 (61.5%) of the 13 patients, and the median gestational week of delivery was 36 (IQR 34–37). The antenatal detection rate was 92.3%, and none of the patients reported postpartum hemorrhage or neonatal death.

### 3.4. Co-Primary Outcome: The Obstetric Outcomes (Type III Versus Other Types)

Two studies included specific information about women with type I and type III VP [48,55]. Of these (*n* = 2), maternal age, rate of ART, rate of diagnosis, GA at diagnosis, rate of emergent cesarean delivery, GA at delivery, and rate of neonatal death were compared between women with type I and type III VP (Figure 4). No studies have compared the outcomes of interest between type II and type III VP.

In this unadjusted pooled meta-analysis comparing type III (*n* = 7) and type I VP (*n* = 17), the mean differences in maternal age (−0.44, 95%CI −6.42–5.53) and gestational age at delivery (0.10, 95%CI −1.69–1.88) were similar. The rates of ART (OR 1.18, 95%CI 0.17–8.17), antenatal diagnosis (OR 0.36, 95%CI 0.01–11.20), and emergent cesarean delivery (OR 0.91, 95%CI 0.08–11.04) were also similar (Table 4).

### 3.5. Co-Secondary Outcome: The Estimated Prevalence of Type III Vasa Previa

The rate of type III VP was examined based on information from five studies [38,48,49,54,55]. Among those (*n* = 5), the rate of type III VP ranged from 1.0% to 35.7% among VP patients. Among women with VP (*n* = 332), the cumulative rate of type III VP (*n* = 19) was 5.7%. The number of pregnant women without VP were available in three studies [48,54,55], and the estimated prevalence of type III VP at delivery ranged from 0.005% to 0.06% (Table 5).

## 4. Discussion

### 4.1. Key Findings

While we need to recognize that only limited information regarding type III VP is available and eligible cases in the present study are limited, the key results of the current study are the following fourfold: (i) obstetric outcomes appear to be similar (type III VP versus type I VP); (ii) among the included studies, type III VP accounts for about 6% of pregnant women with VP; (iii) the characteristics and maternal and neonatal outcomes may be similar (type III VP versus type I VP); (iv) approximately 45% of type III VP was ART pregnancy. Despite limited available data, ART has the potential to be associated with an increased rate of type III VP. To examine the association between ART pregnancy and type III VP, further studies are warranted to reach a robust conclusion.

### 4.2. Strengths and Limitations

The strength of the current systematic review is that it is likely to be the first systematic review focusing on the outcomes and prevalence of type III VP. Although this classification is still uncommon, we believe that knowledge of type III VP may contribute to an increased rate of antenatal diagnosis and improve neonatal outcomes. We also found a high rate of ART pregnancy in women with type III VP. Therefore, ART has the potential to be correlated with a higher rate of type III VP.

However, several key limitations of this study must be recognized. First, most studies were case reports or original articles with small sample sizes, and gross images of the placenta were not available in the original articles; thus, selection bias may need to be considered. Second, the sample size of type III VP was less than 10 in all studies, and the presence of type II errors needs to be recognized. The present systematic review is underpowered to draw robust conclusions regarding the outcomes of type III VP. Nevertheless, we believe that this study will stimulate future research into the outcomes of type III VP.

Third, the classification of type III VP is uncommon, and this problem may have led to a severe selection bias in the systematic literature search. This problem needs to be recognized as a critical limitation of this study. The validity of this classification must be discussed to address this problem. Fourth, careful interpretation is required due to the possible publication bias of present study. For instance, a poor prognosis of a rare type of VP (type III) may be reported more often than the general type of VP (types I and II). Moreover, an original article that focuses on type III VP has the potential to overstate the prevalence of VP since the diagnosis of type III VP is sometimes difficult. In such studies, we considered that the antenatal detection rate of type III VP may have had a severe bias.

Fifth, we attempted to examine the association between ART and the increased rate of type III VP; however, we could not perform a robust examination as no studies have examined the effect of ART on the prevalence of type III VP compared to women with and without VP. One major issue in evaluating the association between ART and type III VP is the absence of a control group. Lastly, the ideal distance between unprotected cord vessels and internal os to diagnose VP is still under debate; however, recent published works suggest that it may be within 2 cm of internal cervical os. Inconsistencies in the definition of VP used among studies and the failure of some studies to define VP at all may have contributed to selection bias in previous research on this topic.

### 4.3. Comparison with Existing Literature

#### 4.3.1. Characteristics, and Obstetric and Neonatal Outcomes of Women with Type III VP

Comparing women who received prenatal diagnoses with those who did not, the neonatal survival rates were 97% and 44%, respectively, and the neonatal transfusion rates were 3.4% and 58.5% [18]. The neonatal outcomes in undiagnosed VP are disastrous, and the rupture of the VP is probably the most catastrophic event for a fetus [7,56].

Antenatal diagnosis is essential to improve neonatal outcomes. In this study, the rate of antenatal diagnosis of type III VP was approximately 85%, whereas no neonatal deaths were observed. The rate of ART pregnancy was approximately 45%, which was high. Nevertheless, this systematic review, which included only 13 women with type III VP, was underpowered to draw robust conclusions. Further studies are warranted to gather information regarding the characteristics and outcomes of women with type III VP.

Although ART has been considered as a risk factor for VP [2,17,57,58,59], comparative studies examining the effect of ART on the rate of pregnancy in women with and without VP are limited [60]. Since ART is associated with an increased rate of VCI [61,62,63,64,65,66,67,68,69,70] and an increased rate of an abnormal placenta (bilobed placenta or succenturiate lobe) [70,71], ART may increase the prevalence of both type I and II VP. As ART pregnancy is high among women with type III VP, ART has the potential to be correlated with a higher incidence of VP.

The present systematic review showed two studies comparing the outcomes of types I and III VP. From the available data, maternal age, ART rate, antenatal diagnosis rate, GA at delivery, emergent cesarean delivery rate, and neonatal mortality rate appeared to be similar. However, because the number of included cases was limited, the presence of type II errors should be recognized.

#### 4.3.2. The Estimated Prevalence of Type III Vasa Previa and Diagnosis Pitfall

The current study has shown that type III VP accounts for approximately 6% of all VP cases. We consider this type of VP is rare, as previous studies have proposed. We hypothesized that type III VP is difficult to diagnose antenatally because types I and II VP are complicated by VCI or placental abnormalities (bilobed placenta or succenturiate lobe). In contrast, type III VP does not have diagnostic characteristics. As previous studies proposed, using color doppler transvaginal ultrasonography, careful examination in both the transverse and sagittal planes is useful to diagnose this rare type of VP [5,40,43].

Nevertheless, antenatal diagnosis of type III VP may be difficult in LMIC. This study [72] did not meet the inclusion criteria of the present study; however, type III VP discovery during labor using simple clinical diagnosis has been reported. In this case report, careful digital examination and amnioscopy enabled the successful detection of VP during labor [72]. Although it is difficult to perform in a clinical setting, careful digital examination for women with the risk factor of VP(ART pregnancy, resolved second-trimester pregnancy, and multiple pregnancies) may help diagnose VP during labor especially in LMIC [1,2,28,59,73,74].

#### 4.3.3. The Mechanism of Developing Type III Vasa Previa

We hypothesized that inadequate decidualization of the placenta is associated with focused atrophy and may lead to development of type III VP [36,37,71]. Previous studies suggested that these changes are more likely to be found in the lower uterine segment due to inadequate blood supply [75,76,77,78]. Unlike VCI and abnormal placenta (bilobed placenta or succenturiate lobe), velamentous vessels in the placenta without a cord or placental abnormalities are understudied. To advance our understanding of the mechanism behind the development of type III VP, it is important to first study this rare type of velamentous vessel.

Our systematic literature search reveals that no basic research has examined the mechanism of type III VP development; thus, the association between ART and type III VP is difficult to discuss, and further studies are warranted to resolve the mechanism of development to type III VP. To aid this problem, the fact that the risk of placenta accreta spectrum is different among the different ART types may be helpful [79].

#### 4.3.4. The Validity of Classifying Vasa Previa into Three Categories

Recent research has found that VP occurs in about 0.3% of pregnancies [11,80,81], while earlier studies have estimated its prevalence at around 0.04% [27,82]. The increased number of ART pregnancies may be more common than previously thought. Types I and II are well-known classifications, whereas type III is uncommon, and only a few authors have proposed this classification [40,48,49]. Moreover, the clinical benefits of type III classification are unknown. Although the clinical benefit is unknown, we believe that type III classification is useful because knowing this rare type of VP may contribute to the antenatal diagnosis of type III VP [5,40,43].

The goal of VP management is to safely prolong the gestational period while avoiding potential complications related to uterine contractions, labor, and rupture of membranes [83]. Previous studies have proposed antenatal hospitalization, and elective cesarean delivery between 34 and 37 weeks of gestation is preferred. In the current national guidelines [18,28,83], management according to the type of VP has not been proposed. The accumulation of detailed clinical information according to the type of VP is necessary to adjust its management.

## 5. Conclusions

### 5.1. Implications for Practice

The characteristics and obstetric and neonatal outcomes of patients with type III VP is understudied. Moreover, this classification is uncommon, and further discussion is required to validate it. While the results of this study did not show a lower detection rate of type III VP, several authors have shown the difficulty of diagnosing it and proposed some tips for its diagnosis. We consider that antenatal diagnosis of this rare type of VP is difficult; thus, knowing this rare type of VP may improve the antenatal diagnosis rate and neonatal outcomes.

The present study showed that ART has the potential to be correlated with a higher rate of type III VP. Numerous studies have shown a positive association between ART and type I VP, and some studies have shown a similar association between ART and type II VP. The association between ART pregnancy and types II and III VP is weak; therefore, future studies are necessary.

The fetal risk is the focus in women with VP. Maternal morbidity may be as high as the co-existence rate of low-lying and placenta previa. Preparing for postpartum hemorrhage (uterotonics, transfusion, uterine compression suture, use of tranexamic acid, and uterine balloon tamponade) may help reduce maternal morbidity [84,85,86,87,88,89]. The maternal morbidity in women with VP is also warranted in future studies.

### 5.2. Implications for Clinical Research

Since previous studies regarding type III VP are scanty, further studies are warranted to examine the outcomes of women with type III VP. If more cases are accumulated and the prognosis of each type of VP is determined, different management strategies may be proposed according to the classifications of VP. Since diagnosing type III VP is difficult, a prospective study may be suitable to determine the true prevalence and outcomes of women with type III VP.

The association between ART and each type of VP is an important topic to be examined. A large-scale retrospective study with multivariate analysis to exclude confounding factors may be preferred to examine this association. All types of VP may be more common in ART pregnancies. Therefore, we may observe that women who conceive on ART still have concerns about VP, even in the absence of placental abnormalities (bilobed placenta or succenturiate lobe).

## Figures and Tables

**Figure 1 biomedicines-11-00152-f001:**
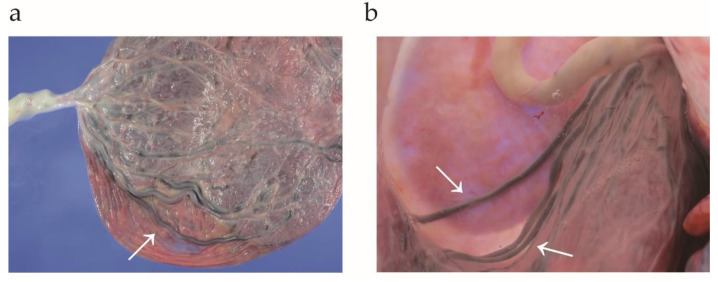
Gross images of type III vasa previa. Reproduced and updated data from Clin Case Rep. 2019; 7: 2263–2264. Hara et al. [39] with copyright permission. (**a**) Marginal cord insertion with an unprotected cord vessel running along the extraplacental membrane (white arrow). (**b**) The placenta and membranes were filled with water to visualize the 3D images of the fetal vessels. The vessels indicated with white arrows were running near the internal cervical os and were diagnosed as vasa previa.

**Figure 2 biomedicines-11-00152-f002:**
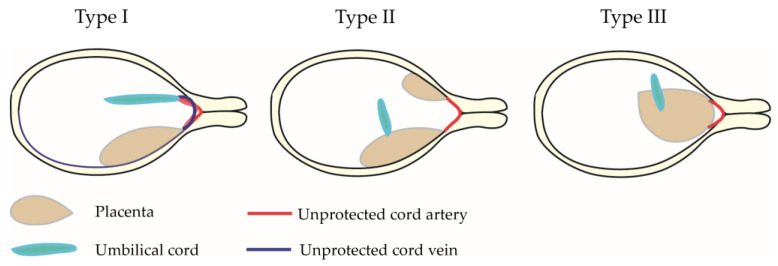
Schema of type I, II, and III vasa previa. The passage of unprotected blood vessels across or near the internal cervical os is common to all three types. Typical characteristics of type I, II, and III vasa previa are shown in the scheme. In general, type I is caused by velamentous cord insertion, type II is caused by a multilobed placenta or placenta with a succenturiate lobe, and type III is caused by a defective placenta and unprotected fetal vessels running outside of the placenta.

**Figure 3 biomedicines-11-00152-f003:**
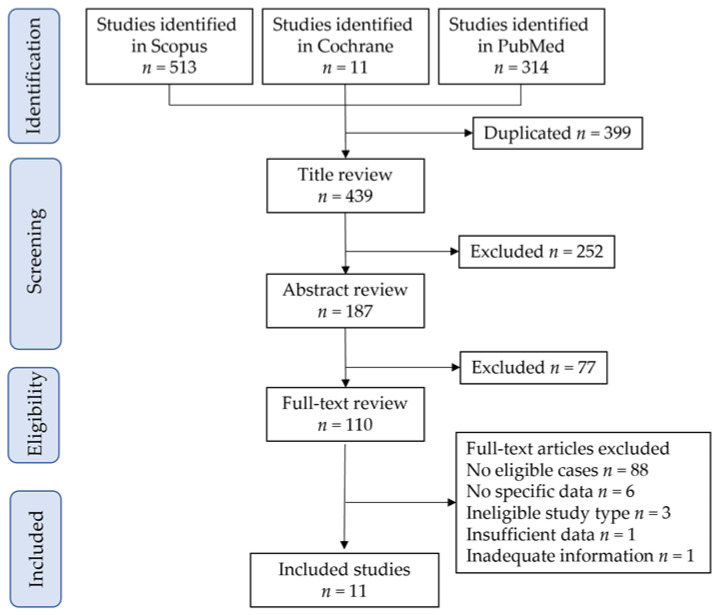
Study selection scheme of the systematic search of previous studies.

**Figure 4 biomedicines-11-00152-f004:**
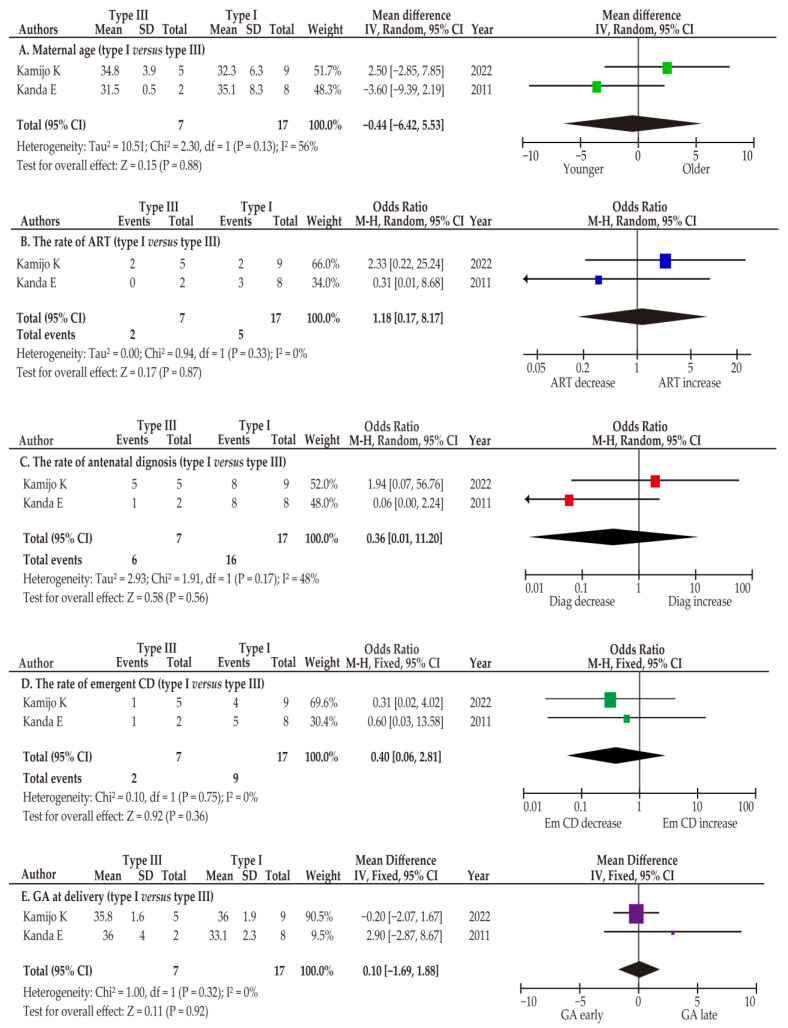
Meta-analysis of the characteristics and outcomes of types I and III vasa previa. Pooled odds ratios for (**A**) maternal age (years), (**B**) the rate of ART pregnancy (unadjusted), (**C**) the rate of antenatal diagnosis (unadjusted), (**D**) the rate of emergent cesarean delivery (unadjusted), and (**E**) gestational age at delivery (weeks) among patients with type I and type III vasa previa. The Forest Plot is ordered by publication year and relative weight (%) of the studies within the strata. The position of colored boxes is a point of the estimated odds ratio and the size represents the weight of study. Heterogeneity among the studies in each analysis was defined as substantial heterogeneity in unadjusted random effect analysis ((**A**): *I*^2^ = 56%), no heterogeneity in unadjusted fixed-effect analysis ((**B**): *I*^2^ = 0%), moderate heterogeneity in unadjusted random effect analysis ((**C**): *I*^2^ = 48%), moderate heterogeneity in unadjusted random effect analysis ((**D**): *I*^2^=32%), and no heterogeneity ((**E**): *I*^2^ = 0%). The above results were calculated using the RevMan ver. 5.4.1. and may vary slightly from their original values. CI, confidence interval; Type III, type III vasa previa; Type I, type I vasa previa; ART, assisted reproductive technology; CD, cesarean delivery; GA, gestational age.

**Table 1 biomedicines-11-00152-t001:** Summary of included studies.

Author	Year	Location	Total	Cont	VP	Type III
Kamijo K [48]	2022	JPN	8723	--	14	5
Tachibana D [49]	2021	JPN	--	--	55	7
Liu N [54]	2021	CHN	79,647	79,486	157	4
Ochiai D [43]	2021	JPN	--	--	1	1
Hata T [42]	2021	JPN	--	--	1	1
Lo A [41]	2020	USA	--	--	1	1
Suekane T [40]	2020	JPN	--	--	1	1
Hara T [39]	2019	JPN	--	--	1	1
Matsuzaki S [5]	2017	JPN	--	--	1	1
Catanzarite V [38]	2016	USA	--	--	96	1
Kanda E [55]	2011	JPN	5131	5121	10	2

Data are presented as numbers. Abbreviations: Year, year of publication; location, location of study; total, total number of included cases; Cont, number of control cases; VP, number of women with vasa previa; Type III, number of women with type III vasa previa; JPN, Japan; CHN, China; USA, United States of America.

**Table 2 biomedicines-11-00152-t002:** Metadata of previous studies examining the obstetric outcomes of women with type III vasa previa.

Author	Year	No.	Age	ART	GA	EmCD	HDP	FGR	PPH	PAS	AD	ND
Kamijo K [48]	2022	5	32	Yes	37	--	--	--	--	--	Yes	--
			39	--	33	Yes	--	--	--	--	Yes	--
			29	--	37	--	--	--	--	--	Yes	--
			39	Yes	35	--	--	--	--	--	Yes	--
			35	--	37	--	--	--	--	--	Yes	--
Ochiai D [43]	2021	1	37	Yes	34	--	--	--	--	--	Yes	--
Hata T [42]	2021	1	38	Unk	37	--	--	--	--	--	Yes	--
Lo A [41]	2020	1	38	Unk	36	--	--	--	--	--	Yes	--
Suekane T [40]	2020	1	32	--	35	--	--	--	--	--	Yes	--
Hara T [39]	2019	1	36	Yes	34	--	--	--	--	--	Yes	--
Matsuzaki S [5]	2017	1	31	Yes	36	--	--	--	--	Yes	--	--
Kanda E [55]	2011	2	32	--	32	Yes	--	--	--	--	Yes	--
			31	--	40	--	--	Yes	--	--	--	--

Data are presented as numbers. Abbreviations: Year, year of publication; No., number of included cases; age, maternal age; ART, assisted reproductive technology; GA, gestational age at delivery; Em CD, emergent cesarean delivery; HDP, hypertensive disorder of pregnancy; FGR, fetal growth restriction; PPH, postpartum hemorrhage; PAS, placenta accreta spectrum; AD, antenatal diagnosis; ND, neonatal death.

**Table 3 biomedicines-11-00152-t003:** Summary statistics for systematic literature review.

Characteristic	(%)
No.	*n* = 13
Age(y)	35 (31.5–38)
ART	5/11 * (45.5%)
Gestational age	36 (34–37)
Antenatal diag	11 (84.6%)
Emergent CD	2 (15.4%)
HDP	0
FGR	1 (7.7%)
PPH	0
PAS	1 (7.7%)
Neonatal death	0

The median (interquartile range) or number (%) is shown. * The method of conception was not clarified in two women. Abbreviations: y, year; ART, assisted reproductive technology; gestational age, gestational age at delivery; diag, diagnosis; CD, cesarean delivery; HDP, hypertensive disorder of pregnancy; FGR, fetal growth restriction; PPH, postpartum hemorrhage; PAS, placenta accreta spectrum.

**Table 4 biomedicines-11-00152-t004:** The comparison of characteristics and outcomes between type I and type III vasa previa.

Author	Year	Type I	Type III	OR or MD (95%CI)
**Age**				
Kamijo K	2022	32.3 ± 6.3	34.8 ± 3.9	2.50 (−2.85–7.85)
Kanda E	2011	35.1 ± 8.3	31.5 ± 0.5	−3.60 (−9.39–2.19)
**ART**				
Kamijo K	2022	2/9 (22.2%)	2/5 (40.0%)	2.33 (0.22–5.24)
Kanda E	2011	3/8 (37.5%)	0/2 (0)	0.31 (0.01–8.68)
**Antenatal diagnosis rate**				
Kamijo K	2022	8/9 (88.9%)	5/5 (100.0%)	1.94 (0.07–56.76)
Kanda E	2011	8/8 (100%)	1/2 (50.0%)	0.06 (0.00–2.24)
**GA at diagnosis**				
Kamijo K	2022	31.8 ± 3.8	31.4 ± 3.7	−0.40 (−4.48–3.68)
Kanda E	2011	26.1 ± 3.9	30	--
**Emergent CD**				
Kamijo K	2022	4/9 (44.4%)	1/5 (20.0%)	0.31 (0.02–4.02)
Kanda E	2011	5/8 (62.5%)	2/2 (100%)	4.09 (0.15–108.94)
**GA at delivery**				
Kamijo K	2022	36.0 ± 1.9	35.8 ± 1.6	−0.20 (−2.07–1.67)
Kanda E	2011	33.1 ± 2.3	36.0 ± 4.0	2.90 (−2.87–8.67)
**ND**				
Kamijo K	2022	0	0	--
Kanda E	2011	0	0	--

Data are presented as the mean ± standard deviation or number (percentage per column). Maternal age (years), ART rate, antenatal diagnosis rate, GA at diagnosis (weeks), emergent CD rate, GA at delivery (weeks) and the rate of neonatal death were compared between women with type I vasa previa and type III vasa previa. Abbreviations: Type I, type I vasa previa; Type III, type III vasa previa; OR, odds ratio; MD, mean difference; CI, confidence interval; ART, assisted reproductive technology; GA, gestational age; CD, cesarean delivery; and ND, neonatal death.

**Table 5 biomedicines-11-00152-t005:** Prevalence of type III vasa previa.

Author	Year	Total	Cont	VP	Type III	Prevalence
Kamijo K [48]	2022	8723	--	14	5 (35.7%)	0.06%
Tachibana D [49]	2021	--	--	55	7 (12.7%)	--
Liu N [54]	2021	79,647	79,486	157	4 (2.5%)	0.005%
Catanzarite V [38]	2016	--	--	96	1 (1.0%)	--
Kanda E [55]	2011	5131	5121	10	2 (20%)	0.04%

Data are presented as numbers (percentages per column). Abbreviations: Total, the total number of cases; Cont, number of control cases; VP, number of vasa previa; Type III, number of women with type III vasa previa; prevalence, the prevalence of type III vasa previa.

## Data Availability

All the data used in this study are published in the literature.

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
