# Peer review of "Current Evidence on Vasa Previa without Velamentous Cord Insertion or Placental Morphological Anomalies (Type III Vasa Previa): Systematic Review and Meta-Analysis"

_biomedicines, 2023, doi:10.3390/biomedicines11010152_

Round 1
Reviewer 1 Report
I read with great interest the paper. I find it well wrote and with good idea research. In my opinion is an important paper suitable for publication after minor revisions.
1. Introduction: Placenta previa has an important impact also in low and middle income countries contributing to high maternal and fetal mortality (see and cite Epidemiology, Outcomes, and Risk Factors for Mortality in Critically Ill Women Admitted to an Obstetric High-Dependency Unit in Sierra Leone. Am J Trop Med Hyg. 2020 Nov;103(5):2142-2148. doi: 10.4269/ajtmh.20-0623. ). In addiction, also can contribute in major risk of infection (see and cite Maternal caesarean section infection (MACSI) in Sierra Leone: a case-control study. Epidemiol Infect. 2020 Feb 27;148:e40. doi: 10.1017/S0950268820000370.)
2. Methods: excellent
3. Results: no comment. Statistical analysis is well performed
4. Discussion: improve the role of antenatal diagnosis to reduce maternal and neonatal mortality and how it is possible also in LMIC. Limitation section is clear. Please discuss also the role of possible causes of placenta previa and the possible action to contrast it also with a rapid recognize of symptom and education of young doctors
5. Conclusion: well done. Please add some proposal that can help the fight against placenta previa and its related mortality.
Author Response
Reviewer #1
I read with great interest the paper. I find it well wrote and with good idea research. In my opinion is an important paper suitable for publication after minor revisions.
Reply:
Thank you for your positive comments. The authors would like to thank the reviewer for his/her constructive critique to improve the manuscript. We have made every effort to address the issues raised and to respond to all comments. Please, find next a detailed, point-by-point response to the reviewer's comments. We hope that our revisions would meet the reviewer’s expectations.
Reviewer #1, comment #1
- Introduction: Placenta previa has an important impact also in low and middle income countries contributing to high maternal and fetal mortality (see and cite Epidemiology, Outcomes, and Risk Factors for Mortality in Critically Ill Women Admitted to an Obstetric High-Dependency Unit in Sierra Leone. Am J Trop Med Hyg. 2020 Nov;103(5):2142-2148. doi: 10.4269/ajtmh.20-0623. ). In addiction, also can contribute in major risk of infection (see and cite Maternal caesarean section infection (MACSI) in Sierra Leone: a case-control study. Epidemiol Infect. 2020 Feb 27;148:e40. doi: 10.1017/S0950268820000370.)
Reply: Lines 49-53
Thank you for your helpful comments. We have cited the suggested studies and have added the statements in the revised manuscript according to the reviewer’s suggestions.
Reviewer #1, comment #2
- Methods: excellent
- Results: no comment. Statistical analysis is well performed
- Discussion: improve the role of antenatal diagnosis to reduce maternal and neonatal mortality and how it is possible also in LMIC. Limitation section is clear.
Reply: Lines 360-367
We would like to thank the reviewer for the helpful comments. According to the reviewers’ suggestion, we have added the discussion regarding the antenatal diagnosis of type III vasa previa in low to middle income country.
Reviewer #1, comment #3
Please discuss also the role of possible causes of placenta previa and the possible action to contrast it also with a rapid recognize of symptom and education of young doctors
- Conclusion: well done. Please add some proposal that can help the fight against placenta previa and its related mortality.
Reply: Lines 421-425
We would like to thank the reviewer for the insightful comments. We completely agree with the reviewer’s opinion. We have added the discussion about maternal morbidity in women with vasa previa. Since the maternal morbidity of women with vasa previa is understudied further studies are warranted and we have added these statements in the revised manuscript.
Reviewer 2 Report
To authors,
The paper is well written. Only one issue that should be pointed out is: whether this theme is so important as being published in this high impact factor journal. I, myself, consider that this theme is worthy of studying but editors will determine it.
I ask you to reconsider the title. Type III vasa previa; although I have been studying the placenta for 4-decades, this is the first time that I have heard of this. This might be due to my short of knowledge. However, I guess that many readers of this journal (including those who just take a glance at the title) may not understand what type III is. Thus, my suggestion is to clarify what type III is in the title.
“Vasa previa without velamentous cord insertion or placental morphological anomalies (type III vasa previa): Current evidence based on systematic review and meta-analysis”. This is my suggestion and you need not use the expression per se.
Author Response
Reviewer #2
Reviewer #2, comment #1
To authors,
The paper is well written. Only one issue that should be pointed out is: whether this theme is so important as being published in this high impact factor journal. I, myself, consider that this theme is worthy of studying but editors will determine it.
I ask you to reconsider the title. Type III vasa previa; although I have been studying the placenta for 4-decades, this is the first time that I have heard of this. This might be due to my short of knowledge. However, I guess that many readers of this journal (including those who just take a glance at the title) may not understand what type III is. Thus, my suggestion is to clarify what type III is in the title.
“Vasa previa without velamentous cord insertion or placental morphological anomalies (type III vasa previa): Current evidence based on systematic review and meta-analysis”. This is my suggestion and you need not use the expression per se.
Reply: Title
We sincerely appreciate the reviewer’s positive comments. According to the reviewer’s suggestion, we have revised the title of manuscript. We trust that the revised manuscript will now be suitable for publication in Biomedicines.